# An Overview on the Synthesis of Fused Pyridocoumarins with Biological Interest

**DOI:** 10.3390/molecules27217256

**Published:** 2022-10-26

**Authors:** Matina D. Douka, Konstantinos E. Litinas

**Affiliations:** Laboratory of Organic Chemistry, Department of Chemistry, Aristotle University of Thessaloniki, 54124 Thessaloniki, Greece

**Keywords:** fused pyridocoumarins, 4-hydroxycoumarins, aminocoumarins, multi-component reactions, [4 + 2] cycloaddition reaction, Povarov reaction

## Abstract

Pyridocoumarins are a class of synthetic and naturally occurring organic compounds with interesting biological activities. This review focuses on the synthetic strategies for the synthesis of pyridocoumarins and presents the biological properties of those compounds. The synthesis involves the formation of the pyridine ring, at first, from a coumarin derivative, such as aminocoumarins, hydroxycoumarins, or other coumarins. The formation of a pyranone moiety follows from an existing pyridine or piperidine or phenol derivative. For the above syntheses, [4 + 2] cycloaddition reactions, multi-component reactions (MCR), as well as metal-catalyzed reactions, are useful. Pyridocoumarins present anti-cancer, anti-HIV, antimalarial, analgesic, antidiabetic, antibacterial, antifungal, anti-inflammatory, and antioxidant activities.

## 1. Introduction

Coumarin derivatives are extensively distributed in plants and trees in nature [1,2,3,4,5,6,7,8]. Natural or synthetic coumarins display a vast array of biological and/or pharmacological activities [9,10,11,12], such as anticoagulant [13,14,15], anti-inflammatory [16,17], anti-HIV and antiviral [18,19,20,21], anticancer [22,23,24,25], antibiotic [26,27,28], antioxidant [29,30,31], antidiabetic [29,32,33], antimicrobial [30,34,35,36], antitubercular [35,37,38], multitarget agents on neurodegenerative diseases [39], etc. Fused coumarins also present biological activity, and many of them, including pyranocoumarins [40], furocoumarins [41,42] and pyrrolocoumarins [41,42], have been isolated from natural sources. Fused pyridocoumarins exhibit varied biological activities, such as antibacterial [43,44,45], antifungal [43,44,45], cytotoxic [46,47], antiproliferative [48], anti-inflammatory [49], analgesic [49], antimalarial [50], antidiabetic [51,52], etc. Some of the fused pyridocoumarins have been extracted from plants. Especially, goniothaline A and goniothaline B (Figure 1) have been isolated from the aerial parts of the Australian rainforest plant *Goniothalamus Australis* and evaluated for in vitro antimalarial activity [53,54,55]. Ganocochliarine F has been isolated from the fruiting bodies of the Chinese fungus *Ganoderma cochlear* and evaluated for inhibition effects on proliferation of fibroblasts NRK-49F [56]. Santiagonamine has been extracted from the stems and branches of *Berberis darwinii* Hook, a South American shrub, and shows wound-healing activity [57]. Polynemoraline C has been isolated from the ethanol extracts of the branches and leaves of *Polyalthia nemoralis* A DC, collected from Hainan province in China [58]. Pharmacokinetic study of polynemoraline C in mouse plasma has been performed for its further preclinical investigation [59]. Schumanniophytine and the isomer isoschumanniophytine have been isolated from the rootbark of *Schumanniophyton magnificum* Harms. (Rubiaceae), a tree found in west central Africa, and possess anti-HIV activity [60,61,62,63]. Even though several reviews referring the chemistry and biological aspects of coumarins and fused coumarins have been published to date, there are few reviews containing topics on the chemistry and biological activity of fused pyridocoumarins [11,42,64,65,66,67]. In this review, we present an overview of the advances described in the literature on the synthesis and biological evaluation of fused pyridocoumarins. The design and synthesis of those derivatives will be presented, accompanied by their biological properties.

## 2. Synthetic Strategies for the Preparation of Fused Pyridocoumarins

The synthesis of fused pyridocoumarins has been achieved by two main routes. One is the formation of a pyridine moiety from a coumarin derivative. The other is the formation of a pyranone moiety from a pyridine or piperidine or phenol derivative.

### 2.1. Pyridine-Ring Formation

The coumarin precursors for the formation of a pyridine ring are aminocoumarins, hydroxycoumarins or other coumarin derivatives.

#### 2.1.1. Synthesis from Aminocoumarins

##### Skraup Reaction

The Skraup reaction is used for the synthesis of quinolines [68,69]. The 3*H*-pyrano [3,2-*f*]quinoline-3-one (**6**) was the first fused pyridocoumarin, prepared in 1919 under the Skraup reaction [70], upon heating of 6-nitrocoumarin (**1**) with glycerol (**2**) in the presence of concentrated sulfuric acid at 145–150 °C and then at 160–170 °C for 6 h in 14% yield [71]. During this reaction, an oxidation of **2** to acrolein (**4**), in parallel to the reduction of **1** to 6-aminocoumarin (**3**), occurred (Figure 1), followed by the addition of **3** to **4**, the formation of the intermediate aldehyde **5**, cyclization of this and dehydration to dihydro pyridocoumarin **6**, which oxidized to give pyridocoumarin **7** [69].

##### Reaction with α,β-Unsaturated Carbonyl Compounds (Skraup–Doebner–von Miller Reaction)

The yield of the above Skraup reaction is relatively low. Doebner and von Miller, by replacing glycerol with α,β-unsaturated ketones in the presence of an acid catalyst, increased the yield of the resulted quinoline derivatives [72,73]. The Skraup–Doebner–von Miller reaction of anilines with 3-substituted α,β-unsaturated carbonyl compounds in the presence of protic acids or Lewis acids resulted mainly in 2-substituted quinolines. Ιntroducing an electron-withdrawing group in the α,β-unsaturated carbonyl (as is the case for 3-subsituted α,β-unsaturated esters), in the presence of TFA, reversed the regioselectivity to give 4-substituted quinolines [73].

In 1994, Heber and Berghaus reported the synthesis of fused pyridocoumarins **11a**–**d** and the azacannabinoidal tetrahydro derivatives **12a**–**d** [74]. The Michael reaction of 4-aminocoumarin moiety of **8a**,**b** with the double bond of arylvinylketone **9a**,**b** gave an intermediate enaminone, which underwent an internal cyclization to form the 1,4-dihydro-adduct **10a**–**d**. The disproportionation of the latter under the applied conditions afforded the mixture of **11a**–**d** (26–36% yield) and **12a**–**d** (23–45% yield) (Figure 2). The reduction of **11a**–**d** with NaBH_3_CN in glacial acetic acid led to the tetrahydropyrido [3,2-*c*]coumarins **12a**–**d** in 70–85% yield.

In 2014, Hamama et al., studied the reactions of 4-aminocoumarin (**13**) with α,β-unsaturated ketones in ethanol/acetic acid (1:1) under reflux [47]. The outcome of those reactions is a regiochemistry reversal to Skraup–Doebner–von Miller reaction (Figure 3). The reaction started from a Michael addition of **13** to α,β-unsaturated ketone **14** to give **A** and **B**, tautomerization of the latter to **C**, followed by cyclization of **C** and removal of water to dihydropyridocoumarin **D**, which by oxidation led to pyridocoumarin **15** in 58% yield. The synthesis of **17** was achieved in 76% yield by the reaction of **13** with dibenzylideneacetone (**16**). The similar reaction of 2,6-dibenzylidenecyclohexanone (**18**) resulted in pyridocoumarin **19** in 73% yield. The new compounds were tested for their antitumor activity in vitro against Ehrlich ascites carcinoma cells (EAC) and were found to be three times more toxic than 5-fluorouracile 5-FU.

In 2017, Samanta and coworkers reported the Cu(OTf)_2_ (10 mol%)-catalyzed synthesis of fused pyridocoumarins **22a**–**z** from aminocoumarins **13**, **23a**–**d** and β,γ-unsaturated α-ketoesters **20a**–**l** under solvent-free conditions, microwave irradiation and open atmosphere [75]. The reaction mechanism was similar with the above applied. A Michael addition of 4-aminocoumarin (**13**) to the Lewis acid, Cu(Otf)_2_, activated **20**′, led to the Michael adduct **C** (Figure 4). The 1,4-dihydropyridocoumarin **21** was formed by the cyclization of the latter and water elimination. Oxidation of **21** in the presence of Cu(Otf)_2_ under the reaction conditions afforded the pyridocoumarin **22a**. It must be mentioned that the yield of this conversion was only 19% without the presence of the catalyst.

The following year, Adib et al., synthesized a series of fused pyridocoumarins **26a**–**p** by the reactions of 4-aminocoumarins **13**, **24a** (prepared in situ from 4-hydroxycoumarin and ammonium acetate) with α-azidolactones **25a**–**j** in the presence of NaOH under heating at 60 °C for 20 min [52]. According to the mechanism proposed (Figure 5), the aminocoumarin **13** deprotonated by NaOH and the conjugate base **A** added in a Michael addition to the α-azidolactone **25a** to give the intermediate **B** under removal of a nitrogen molecule. An imine–enamine tautomerization of the latter followed by cyclization led to tricyclic intermediate **C**, which by water elimination and tautomerization of imine resulted in amino-substituted pyridocoumarin **26a**. The synthesized compounds were evaluated for their α-glucosidase inhibitory activity and exhibited in vitro yeast α-glucosidase inhibition with IC_50_ = 101.0–227.3 μM, better than the standard drug acarbose (IC_50_ = 750.0 μM). Compound **26i** was the most potent.

In 2019, Osyanin and coworkers received regioselectively the 3-substituted pyridocoumarins **28a**–**I** by the reaction of 4-aminocoumarin (**13**), with the β-formyl substituted 4*H*-chromenes **27a**–**i** in 52–74% yield [76]. The reaction proceeds through a Michael addition of **13** to the α-carbon of chromene carbaldehyde, e.g., **27i**, leading to the Michael adduct **A**, according to their former work [77]. The condensation of the latter under cyclization possibly led to dihydropyridine **B** (Figure 6). Aromatization of the pyridine ring under opening of the pyran ring furnished the final product **28i**.

##### Povarov Reaction

A Povarov reaction is a Diels–Alder reaction between an *N*-aryl imine and an electron-rich dienophile in the presence of Lewis acid as catalyst, used for the synthesis of quinolines [78,79,80,81,82]. The reaction is an inverse electron demand Diels–Alder (IEDDA). The one-pot synthesis using aromatic amine, aldehyde, and electron-rich alkene as a MCR is an advance of the Povarov reaction, leading to quinolines [83,84].

In 2008, Bodwell and coworkers prepared the 1,2,3,4-tetrahydopyridocoumarins **33a**,**b** (36:64) from the Povarov reaction of imine **31**, synthesized by the reaction of 3-aminocoumarin (**29**) with p-nitrobenzaldehyde (**30**), and the 3,4-dihydro-2*H*-pyran (**32**) in the presence of Yb(OTf)_3_ as a catalyst (Figure 7). Similar reactions of **31** with various electron-rich dienophiles resulted in the corresponding tetrahydropyridocoumarins in good yields and variable diastereomeric ratios [85]. They also prepared some of the products, such as **33a**,**b**, by the one-pot three component version of this reaction. The reaction proceeds by an IEDDA [4 + 2] cycloaddition reaction of alkene to the imine **A**, catalyzed by Yb(OTf)_3_, to **B**, which upon tautomerization gave the products **34**. Oxidation of these products led to the pyridocoumarins, e.g., **35** resulted in **36** by oxidation with bromine.

The same group, in 2011, synthesized the fused pyridocoumarins **40**, **45** by the intramolecular Povarov reaction of 3-aminocoumarin (**29**) and the unsaturated ethers of salicylaldehyde **37**, **42**, respectively, in the presence of 5 mol% Yb(OTf)_3_ [86]. In the case of **37** the *cis*-tetrahydropyridocoumarin **39** was isolated along with the reduction product **41** of the intermediate imine **38**. The reactions of 3-aminocoumarin derivatives with *O*-cinnamylsalicylaldehydes, e.g., **46**, resulted in the *trans*,*trans*-tetrahydropyridocoumarins, such as **47**, in the presence of 10 mol% Yb(OTf)_3_ at r.t. (Figure 8). In the case of *N*-cinnamyl-2-pyrrolcarbaldehyde **48,** the *cis*,*trans*-adduct **49** with [5,6] fused ring system was obtained.

In 2015, Khan and coworkers utilized the 10 mol% Yb(OTf)_3_ in an one-pot three component Povarov reaction of 3-aminocoumarins, aldehydes and 5,6-unsubstituted 1,4-dihydropyridine derivatives [87]. The products, hexahydro-1H-chromeno [3,4-*h*] [1,6]naphthyridine-3-carboxylate derivatives **53a**–**y**, isolated in 72–91% yield, have the *exo*-conformation, as is suggested by ^1^H-NMR spectra and X-ray diffraction analysis of compound **53a** (Figure 9).

The same group, in 2012, prepared the pyrido [2,3-*c*]coumarin derivatives **36**, **55a**–**n** by the molecular iodine catalyzed one-pot Povarov reaction of 3-aminocoumarins **29**, **50a**,**c** with aromatic aldehydes **30**, **51a**–**c**,**e**–**h**,**m** and alkynes **54**, **56a**–**e** [88]. The condensation reaction of **29** with **51a** led to the formation of imine **A**. The [4 + 2] Povarov reaction of **A** with alkyne **56** gave 1,4-dihydropyridocoumarin **B**, which upon tautomerization to **C** and oxidation by air resulted in the pyridocoumarin **55** (Figure 10).

In 2011, Majumdar and coworkers used the BF_3_.Et_2_O (10 mol%) as a catalyst for the Povarov three-component reaction of 6-aminocoumarin (**57a**) with aromatic aldehydes **51** and phenylacetylene (**54**) to prepare the angular pyrido [3,2-*f*]coumarins **58a**–**c** [89]. The similar reaction of 7-amino-4-methylcoumarin (**59a**) with the anisaldehyde **51c** and **54** resulted in linear pyridocoumarin **60** (Figure 11). The intermediate imine **A** underwent a Diels–Alder reaction with **54** to give the dihydropyridine **B**. Tautomerization of the latter to **C** followed by oxidation by the air afforded the final product **58**.

In 2013, our group reported the synthesis of fused pyridocoumarins **62a**–**f**, **63a**,**b** and **64** under a three component Povarov-type reaction of 6- or 7-aminocoumarins **57a**–**f**, **59a**,**b** with *n*-butyl vinyl ether (**61**) in the presence of 10 mol% molecular iodine [90]. Iodine, a mild Lewis acid, catalyzes the reaction of **61** with the aminocoumarin **57b** for the formation of the intermediate imine **A** (Figure 12). The intermediate **B** is formed by an aza-Diels–Alder reaction of **A** with a second molecule of **61**, under iodine catalysis, and tautomerization. Elimination of *n*-butanol resulted in 1,2-dihydropyridocoumarin **C**, which upon oxidation led to the final product **62b**.

In 2014, Ganguli and Chandra synthesized the [5,6]- and [6,7]-fused pyridocoumarins **66a**–**i**, **67a**–**f**, respectively, under an iodine-catalyzed three-component reaction of 6-aminocoumarin (**57a**), aldehydes and styrene (**65**) in aqueous micellar conditions in the presence of sodium dodecyl sulfate (SDS) [91]. During this process, the 6-benzylaminocoumarins **68a**–**i** were isolated. The intermediate aldimine **A** reacted with **65** in an inverse electron demand Diels–Alder reaction to give the Povarov adducts **B** and **C** (Figure 13). These adducts aromatized to pyridocoumarins **66** and **67**, respectively, by a hydrogen transfer to aldimine **A**, which gave the benzylaminocoumarin **68**.

In 2014, our group used FeCl_3_ as a catalyst for the three component domino reactions of 6- or 7-aminocoumarins **57** or **59** with benzaldehyde (**51a**) and phenylacetylene (**54**). The reactions were performed in toluene under reflux or under microwave irradiation at 170 °C in the presence of air or p-benzoquinone leading to the synthesis of 2,4-diphenyl-substituted fused pyridocoumarins **66**, **67**, **69** or **71**, **72** [92]. The intermediate imine **A** underwent nucleophilic addition from alkynylated complex **B** to give propargylamine complex **C**. The **C,** through intramolecular arylation afforded the vinyl complex **D**, which on decomposition resulted in 1,4-dihydropyridocoumarin **E**. Tautomerization of the latter and oxidation by air led to pyridocoumarin **66a** (Figure 14). We had tested the new compounds as inhibitors of lipid peroxidation. Compound **66a**, **67a**, **69b** presented 100% inhibition of antilipid peroxidation at 0.1 mM.

The same year, Khan and coworkers synthesized furo- and pyrano-tetrahydropyrido [2,3-*c*]coumarin derivatives by the one-pot three-component reactions of 3-aminocoumarin (**29**) with aromatic aldehydes **30**, **51** and 2,3-dihydrofuran or 3,4-dihydropyran (**32**) in the presence of Fe_2_(SO_4_)_3_.xH_2_O in refluxing acetonitrile [93]. The reactions with 3,4-dihydropyran (**32**) resulted in tetrahydropyridocoumarins **74a**–**j** and **75a**–**j** as *endo*–*exo* and *endo*–*endo* diastereomeric products, respectively (Figure 15). The **74** were the major products, while the **75** were the minor products, as established by the coupling constants of the ^1^H-NMR spectra. The XRD crystallographic data of **74e** revealed the endo–exo configuration. From the performed docking studies, it was found that most of the derivatives **75** present inhibition activity against human dopamine D3 receptor. The blockage of this receptor is effective for potential pharmacotherapy of several neuropsychiatric disorders.

In 2015, the same group reported an intramolecular Povarov reaction of 2-propargyloxybenzaldehydes **42**, **76a**–**d** with 3-aminocoumarins **29**, **50a**–**f** catalyzed by triflic acid (10 mol%) in acetonitrile under reflux for the synthesis of fused pyridocoumarin derivatives **45**, **77a**–**p** (Figure 16). The structure of **45** and **77m** was confirmed by X-ray diffraction analysis [94]. The plausible mechanism for this reaction is similar to the mechanism proposed in Figure 7 with formation of intermediate imine, intramolecular Diels–Alder reaction, tautomerization and aromatization through air oxidation.

In 2015, Xi and Liu synthesized ferrocenyl-substituted pyrido [3,2-*g*]coumarins **80a**–**o** by Povarov three-component reaction of 7-amino-4-methylcoumarin (**59a**) with aromatic aldehydes **51**,**78** and ferrocenylacetylene (**79**) in the presence of Ce(OTf)_3_ as a catalyst [95] under refluxing toluene (Figure 17). They studied the antioxidant activity of the ferrocenylcoumarin derivatives and it was found that these compounds can trap radicals and inhibit DNA oxidation. Derivatives with electron-donating group at 8-position, such as **80d**, **80n**, **80o**, possess higher inhibitory effect on AAPH-induced oxidation of DNA.

In 2016, Gurumurthy et al., synthesized tetrahydropyrido [2,3-*c*]coumarin derivatives **74** and dihydropyrido [2,3-*c*]coumarin derivatives **82a**,**b**, **84a**–**c** by the one-pot three component Povarov reaction of 3-aminocoumarin (**29**) with aromatic aldehydes **30**, **51**, **78** and 3,4-dihydropyran (**32**) or 2-vinylnaphthalene (**81**) or diethylacetylenedicarboxylate (**83**) under BiCl_3_ catalysis in acetonitrile at room temperature [96]. The reactions gave, stereoselectively, the *endo*–*exo* products **74**, as it was established by ^1^H-NMR experiments. The similar reaction with diisopropyldiazadicarboxylate (**85**) resulted in the fused triazinocoumarin derivatives **86a**–**d**. A stepwise mechanism has been proposed for this reaction (Figure 18). By the electrophilic interaction of 3,4-dihydropyran (**32**) to the intermediate imine **A**, activated as **B** by the Lewis acid, BiCl_3_, the intermediate **C** was formed. The latter underwent a ring closure in *anti*-mode via an intermolecular attack by the carbon-4 of coumarin ring to give the *endo*–*exo* product **74a**. The synthesized compounds were evaluated for their antioxidant activity, determined by the DPPH radical scavenging activity. Compounds **84b** and **86a** exhibited good free radical scavenging activity, but lower than the reference compounds α-tocopherol and butylated hydroxytoluene (BHT).

In 2016, Chen et al., reported the synthesis of substituted pyrido [2,3-c]coumarins **55a**, **88a**–**x** by a one-pot three-component reaction of acetophenones (mainly), aromatic aldehydes and 3-aminocoumarin (**29**) in the presence of equimolar amount methanesulfonic acid in refluxing acetonitrile [97]. As a plausible mechanism, they proposed the addition of enol **87a**′, formed in the presence of acid by the tautomerization of acetophenone **87a**, to the intermediate imine **A**, the condensation product from **29** and benzaldehyde **51a**, in an asynchronous [4 + 2] cycloaddition reaction. Subsequently, the coumarin ring of **B** added to the ketone carbonyl to give the intermediate **C**. The latter by tautomerization, elimination of water and oxidation under air resulted in the product **55a** (Figure 19).

##### Friedlander Reaction

Friedlander reaction is the reaction of *o*-aminobenzaldehydes with carbonyl compounds containing *α*-methylene group in the presence of base or acid, or without catalyst under heating, leading to the synthesis of quinolines [98,99,100]. For the mechanism of this reaction two routes are accepted, Schiff base formation or intermolecular aldol reaction. In both cases, a cyclodehydration follows to give quinoline.

In 2013, Siddiqui and Khan applied the Friedlander reaction of 4-amino-3-formylcoumarin (**89**) with active methylene carbonyl compounds **90a**–**m** or malononitrile (**91**) under solvent-free conditions at 80 °C in the presence of chitosan as a green catalyst to get the fused pyrido [3,2-*c*]coumarin **92a**–**n** [101]. Barbituric acid (**90b**), Meldrum’s acid (**90e**), 1,3-indandione (**90f**), dimedone (**90g**), 4-hydroxycoumarin (**90h**), ethyl acetoacetate (**90k**), acetylacetone (**90l**) were between the active methylene carbonyls. As a plausible mechanism the chitosan abstracted a hydrogen to give carbanion **A**, which added to the electrophilic carbon of formyl-group of **B** (Figure 20). Elimination of water from the resulted specie **C** led to unsaturated coumarin intermediate **D**. A 1,2-Addition of the 4-NH_2_ group to the carbonyl of **D** and elimination of water from the intermediate **E** afforded the product **92**.

##### From Propargylaminocoumarins

In 2007, Lee and coworkers synthesized 9,10-Di-*O*-camphanoyl-4,8,8-trimethyl-7,8,9,10-tetrahydro-2*H*-pyrano [2,3-*f*]quinolin-2-one (**97**) by the esterification of aza-*cis*-khellactone **96** with (*S*)-(-)-camphanoyl chloride [102]. The latter was prepared by the asymmetric Sharpless dihydroxylation with AD-mix-α of 4-methyl-1′-azaseselin (**95**), which was formed by the aza-Claisen rearrangement and cyclization of propargylaminocoumarin **94** in the presence of CuCl in refluxing THF (Figure 21). The substitution of 3-chlorobutyne **93** by the aminocoumarin **59a** resulted in adduct **94**. Compound **97** as well as analog pyran derivatives have been studied for their ant-HIV activity using the HIV-1 IIIB strain in H9 lymphocytes. It was found that **97** has an anti-HIV activity with EC_50_ = 0.77 μM and therapeutic index (TI) > 42.

In 2011, Majumdar and coworkers used iodine for the Claisen rearrangement and cyclization of 6-propargylaminocoumarins **102a**–**c** and **105** to obtain selectively the angular dihydropyridocoumarins **103a**–**c** and the pyridocoumarin **106**, respectively [103]. For the mechanism, they suggested an initial formation of the iodonium intermediate **A**, followed by the nucleophilic attack of the 5-carbon of aromatic ring to the activated triple bond. Hydrogen abstraction by the base from the intermediate **B** led to the final product **103** (Figure 22). In the case of non-*N*-substituted dihydro-intermediate **C** the oxidation by the iodine might be responsible for the synthesis of pyridocoumarin **106**.

The same year, our group using BF_3_.Et_2_O under microwave irradiation obtained, also selectively, the angular [5,6]-fused pyridocoumarins **108a**,**b** through the aza-Claisen rearrangement of propargylaminocoumarins **107a**,**b** [104]. The pyridocoumarins **110a**,**b** were isolated similarly from the propargylaminocoumarins **109a**–**c**. The imino-adduct **A** was formed through the aza-Claisen rearrangement of **107a**, followed by tautomerization to **B** (Figure 23). 1,5-H Shift of the latter gave the intermediate **C**, which by a Diels–Alder intramolecular reaction furnished the 1,2-dihydropyridocoumarin **D**. Oxidation of **D** led to the final product **108a**.

In 2013, our group utilized the Au-NPs for the catalyzed synthesis of the pyridocoumarins **108a**,**b** and **110a**,**c**–**e** in excellent yields from the propargylaminocoumarins **107a**,**b** and **109a**–**d**, respectively [105]. A plausible mechanism with electrophilic aromatic substitution of the benzene ring of coumarin with the activated alkyne-π complex of **A** to the vinyl-Au intermediate **B** through a 6-*endo*-*dig* cyclization is presented in Figure 24. 1,3-H Shift under regeneration of the catalyst gave 1,2-dihydropyridocoumarin **C**, which by air-oxidation resulted in the isolation of [5,6]-fused pyridocoumarin.

In 2014 Majumdar and Ponra synthesized the dihydropyrido [3,2-*f*]coumarins **111d**–**j** from the propargylaminocoumarins **102d**–**j** in the presence of FeCl_3_ [106]. The expected pyrido [3,2-*f*]coumarins were not isolated during the above reactions. For the proposed mechanism, FeCl_3_ activates the alkyne moiety of **102** to give the intermediate π-complex **A** (Figure 25). An intramolecular 6-*endo*-*dig* cyclization of **A** produced the charged species **B**. Upon deprotonation of the latter, followed by an 1,3-H shift and elimination of FeCl_3_ the final product **111** was formed.

In 2019, Han and coworkers prepared goniothaline A (**118**) through the Ag-catalyzed cycloisomerization of propargylaminocoumarin **117**, while by the following regioselective demethylation due to neighboring pyridine nitrogen goniothaline B (**119**) was obtained [53]. Propargylamino-compound **117** has been synthesized by propargylation of aminocoumarin **116**. which has been prepared via a five-step procedure starting from 2,5-dihydroxybenzaldehyde (**78e**) (Figure 26).

In continuation of their work, the same group reported the synthesis of pyrido [3,2-*c*]coumarins **124a**–**i** by the AgNO_3_ catalyzed cycloisomerization of 4-propargylaminocoumarins **123a**–**i** [107]. Propargylaminocoumarins have been synthesized by the nucleophilic substitution of 4-chlorocoumarins **121a**–**g** with the propargylamine salts **122a**–**c** (Figure 27). Polynemoraline C (**124i**) is a natural product synthesized by this method. As a plausible mechanism, the alkyne moiety of **123a** coordinated with silver catalyst to give intermediate **A**. An intramolecular attack of the enamine carbon atom to the electrophilic alkyne bond of **A** via a 6-*endo*-*dig* cyclization resulted in the 6-membered **B**. 1,3-H Shift under demetallation gave the 1,2-dihydro pyridocoumarin **C**, which oxidized to afford the final product **124a**.

Very recently, our group synthesized bis-fused pyridopyranocoumarins **128a**,**b**, **131**, **134a**,**b** from the propargylaminocoumarin derivatives **126a**,**b**, **127a**,**b**, **120**, **133a**,**b** in excellent yields under Au-NPs catalyzed cycloisomerization reaction followed by air oxidation [108]. The propargylaminocoumarins have been synthesized from aminohydroxycoumarins **125a**,**b** and **129** under propargylation with propargyl bromide (**99**) or 3-chloro-3-methylbutyne (**93**) (Figure 28). The compounds were tested for their antioxidant and anti-AChE activities. The derivatives **128a**, **132a**, **134a**,**b** presented promising anti-lipid peroxidation and anti-AChE activities.

##### Multi Component Reactions (MCR) of Aminocoumarin

Multicomponent reactions (MCRs) are an important method for the one-pot synthesis of organic compounds under atom economy of the three or more participating starting materials [109,110,111,112,113]. Povarov reaction, as we have mentioned earlier, is an application of MCRs for the synthesis of pyridocoumarins.

In 2012, Khan and Das utilized the MCR of 3-aminocoumarins **29**, **50a**,**b**, aldehydes **30**, **51**, **78** and cyclic 1,3-diketones **90m**, **135a**,**b** to prepare 1,4-dihydropyrido[2,3-*c*]coumarin derivatives **136a**–**s** in the presence of catalytic amount (20 mol%) of *p*-toluenesulfonic acid (*p*-TSA) [114]. The Knoevenagel condensation of benzaldehyde (**51a**) with dimedone (**90g**) gave the adduct **A**, which underwent a Michael addition of 3-aminocoumarin (**29**) leading to the intermediate **C** (Figure 29). An intramolecular ring closure reaction of the latter followed by dehydration of the intermediate **D** resulted in the final product **136a**.

The same year, Paul and Das via a MCR of 4-aminocoumarin (**13**), aromatic aldehydes **30**, **51** and indandione (**90f**) or dimedone (**90g**) in the presence of the organic catalyst, (±)-lactic acid, in a green solvent, ethyl-*L*-lactate, obtained the 1,4-dihydropyrido [3,2-*c*]coumarin derivatives **137a**–**h** and **138a**–**h**, respectively [115]. At first, a Knoevenagel condensation led to the intermediate **A**. The formation of a H-bond between lactic acid and the carbonyls of **A** may increase the electrophilicity of the carbonyls, accelerating the Michael addition of 4-aminocoumarin (**13**) (Figure 30). The intermediate **B** formed, upon tautomerization to adduct **C** and elimination of water, resulted in the final products **137** or **138**.

In 2013, the same group prepared the pyrido [3,2-*c*]coumarin derivatives **139a**–**d** by a MCR domino process from 4-aminocoumarin (**13**), aldehydes and malononitrile (**91**) under heating in water at 80 °C in the presence of triethanolamine, a Lewis-base-surfactant combined catalyst (LBSC), which acted as a catalyst to activate the substrates and as a surfactant forming colloidal particles [116]. A Knoevenagel condensation of aldehyde **51a** with **91** gave the α,β-unsaturated nitrile **A**, which upon Michael addition of 4-aminocoumarin (**13**) afforded the intermediate **B** (Figure 31). Tautomerization of the latter to **C** followed by intramolecular cyclization, catalyzed by triethanolamine, led to the intermediate **D**. Air oxidation resulted in the final product **139a**.

In 2014, Kar and coworkers reported the synthesis of pyranoquinolin-3-ones **142a**,**b**, **143a**,**b**, **149** and pyranoacridin-3-ones **146a**–**c** by the thermolysis of enamino imine hydrochloride derivatives of 7-aminocoumarin **141a**,**b**, **148** and **145a**–**c**, respectively [117]. These derivatives were prepared by the three-component reaction of 7-aminocoumarin (**57a**) with β-chloroacrolein derivatives **140a**,**b** or 1-chloro-3,4-dihydronaphthalene-2-crabaldehydes **144a**–**c** or 2-chloroacenaphthylene-1-carbaldehyde **147** (Figure 32). Oxidation of compounds **146a**–**c** with DDQ resulted in the fully aromatic 3*H*-benzo[*h*]pyrano [3,2-*a*]acridine-3-one derivatives **150a**–**c**. A few years later, in 2017, Patra performed a modification of the above three-component reaction by using only one equivalent of 7-aminocoumarin (**57a**) and β-chloro-α,β-unsaturated aldehydes in methanol at 15 °C without the presence of hydrochloric acid [118]. This procedure led to the preparation of the intermediate chlorovinyl imine derivatives **151a**,**b** and **152a**–**d** (Figure 33). Thermolysis of the latter at 230–260 °C resulted in the synthesis of pyranoquinoline-3-one derivatives **142a**,**b** and pyranoacricine-3-one derivatives **146a**–**d** with better yields.

In 2017, Kausar and Das reported the synthesis of pyrido [2,3-*c*]coumarin derivatives using the three-component reaction of 3-aminocoumarins, aldehydes and phenylacetylene in the presence of CuI-Zn(OAc)_2_ as catalyst without solvent, under ball milling green conditions [119]. The reaction involves one C-N coupling and two C-C couplings in a combo–catalysis cycle (Figure 34) in the proposed mechanism. The activation of terminal alkyne produced the copper carbide **A**. By a C-C and C-N bond formation between aminocoumarin (**29**), benzaldehyde (**51a**) and carbide **A** under the Lewis acid Zn(OAc)_2_ catalysis the propargylamino intermediate **B** was formed. Oxidative C-H insertion after the activation of triple bond, and the π-bond of **B** by CuI, led to the seven-membered intermediate **C**, which underwent a C–C coupling under reductive elimination to give CuI and the 1,2-dihydropyrido [3,2-*c*]coumarin **D**. Oxidation of the latter resulted in the final product **55a**.

Recently, Rad-Moghadam and coworkers obtained the spiro 1,4-dihydropyrido [3,2-*c*]coumarin derivatives **156a**–**f** via a three-component reaction of 4-aminocoumarin (**13**), isatin derivatives **154a**–**f** and 3-methyl-1*H*-pyrazol-5-amine (**155**) in the presence of *p*-TSA in ethanol under sonication or heating [120]. They proposed two possible pathways (Figure 35). In each path, A or B condensation of isatin (**154a**) with **13** or **155** led to the formation of the corresponding intermediates **A** or **E**. Nucleophilic addition of **155** or **13** to the **A** or **E**, respectively, gave the adducts **B** or **F**. The [3,3]-Sigmatropic rearrangement to **C** or **G**, followed by elimination of ammonia, afforded **D** or **H**. Tautomerization of both intermediates resulted in the final spiro compound **156a.**

The same year, Bregadiolli and coworkers synthesized chromeno [4,3-*b*]pyridine derivatives **158a**–**i** from a MCR between 4-aminocoumarin (**13**), aromatic benzaldehydes and ethyl benzoylacetate (**157**) catalyzed by NbCl_5_ [121]. In the proposed mechanism, a Knoevenagel condensation occurred between the enolic form of ethyl benzoylacetate (**157**) and benzaldehyde complexed with niobium, leading to the unsaturated intermediate **B** (Figure 36). The complexation of NbCl_5_ with carbonyl-oxygen of **C** reduced the electron density of the double bond facilitating the attack of NH_2_-group of **13** to give the intermediate **D**. After removing of a proton from the ammonium group and cyclization of the resulting **E** under elimination of water to **F**, followed by deprotonation, the 1,4-dihydropyrido [3,2-*c*]coumarin **158a** was formed.

##### Metal-Catalyzed Reactions of Aminocoumarin Derivatives

In 2008, Majumdar and coworkers utilized the Pd-catalyzed intramolecular Heck reaction of 6- or 7-benzoylaminocoumarins **162a**–**c** or **164a**–**c** for the regioselective synthesis of angular 3*H*-pyrano [3,2-*a*]phenanthridine-3,8(7*H*)-diones **165a**–**c** or linear 11-methyl-5*H*-pyrano [3,2-*b*]phenanthridine-5,9(6*H*)-dione **166a**–**c**, respectively [122]. In the case of no *N*-substituted amides **162a** and **164a** the Ag_2_CO_3_ was used as a base in the place of KOAc at 160 °C (Figure 37).

In 2012, Majumdar et al., reported the synthesis of pyrido [3,2-*f*]coumarins **168a**–**f** by the Indium (III) chloride-catalyzed reaction of 5-allyl-6-aminocoumarin (**167**) with benzaldehydes **51** [123]. In the proposed mechanism, the imine **A** formed from **167** and **51a** in the presence of InCl_3_ at first, followed by an 1,5-H shift to the intermediate **B**. The aromatization of benzene ring to **C** via an 1,7-H shift followed by an 6π-electrocyclization to intermediate **D** and a subsequent 1,5-H shift to **E**, which was oxidized, led to the final product **168a** (Figure 38).

In 2017, Nath, a coworker of Majumdar, extended the former [66] regioselective Pd-catalyzed synthesis of linear 11-methyl-5*H*-pyrano [3,2-*b*]phenanthridine-5,9(6*H*)-diones **166a**–**g** using Cs_2_CO_3_ as a base at lower temperature, 95 °C for 6 h [124]. In the case of amidocoumarin **164a,** the base was a mixture of Ag_2_CO_3_ (2 equivalents) and Cs_2_CO_3_ (2 equivalents) and the intramolecular Heck reaction performed at the elevated temperature of 120 °C (Figure 39).

In 2017 also, Xie, Su and coworkers synthesized 6*H*-chromeno [4,3-*b*]quinoline-6-ones **170a**–**t** by a copper-catalyzed cyclization of 4-arylaminocoumarins **169a**–**t** using the *N*-methyl moiety of DMF as the source of methine group [125]. They tested *N*,*N*-dimethylacetamide and *N*,*N*-dimethylaniline as a possible source of methine moiety, obtaining low yields of the product, while *N*,*N*-diethylformamide did not give any conversion. A possible mechanism has been proposed with addition of 4-phenylaminocoumarin (**169a**) to the iminium salt **A**, generated from DMF (Figure 40). The intermediate **B** formed after elimination of MeNHCH=O gave the α,β-unsaturated imine **D**, which upon attack from NaHSO_3_ to adduct **E** followed by intramolecular cyclization generated the dihydropyridine intermediate **F** (path A). Another possibility is the 6π electrocyclization of **D** to afford intermediate **F** (path B). Oxidation, next, of **F** led to the final product **170a**.

Recently, Ackermann and coworkers utilized 4-arylaminocoumarins **169**, **171** for the synthesis of 6*H*-chromeno [4,3-*b*]quinoline-6-ones **170**, **172** through electro-oxidative cyclization in the presence of DMF as a methine source in a glassy carbon (GC) anode and a platinum (Pt) cathode [126]. In the proposed mechanism, iodine radicals, generated anodically, afforded intermediate **A** from **169a**, which is converted to radical **B**, releasing iodine anion (Figure 41). The iminium intermediate **C**, formed by anodic oxidation of DMF, reacted as an electrophile with **B** to give intermediate **D**. Elimination of MeNHCHO furnished the imine **E**, which was attacked by NaHSO_3_ to afford intermediate **F**. Intramolecular cyclization to **G**, followed by oxidation, led to the final product **170a**.

In 2019, Samanta, Kumar and coworkers reported the synthesis of substituted chromeno [4,3-*b*]pyridines **174a**–**p** from 4-aminocoumarins **13**, **23a**–**d** and α-alkynyl-β-aryl nitroolefins **173a**–**e** in the presence of copper acetate by heating in 2-methyltetrahydrofuran, as a green solvent, under aerobic conditions [127]. This reaction is a domino protocol via a [3 + 3] annulation reaction promoted by Cu(OAc)_2_, as is suggested by the authors (Figure 42). The compounds were tested against CAG repeat RNAs that cause Hantington’s disease. Derivatives **174c** and **174o** presented higher affinity (nanomolar) and selectivity for diseased r(CAG)^exp^ RNA compared to regular duplex AU-paired RNA.

#### 2.1.2. Synthesis from Hydroxycoumarins

##### Multi Component Reactions (MCR) of Hydroxycoumarins

In 2004, Kidwai et al. synthesized the bis(benzopyrano) fused 1,4-dihydropyridines **176a**–**f** by a three-component reaction of 4-hydroxycoumarin (**120a**), aromatic aldehydes **51**, **78** and ammonium acetate under one-step procedure using acidic alumina and silica gel as solid support under MW irradiation (Method A) [128]. This method was faster and had better yields than the two-step procedure (Method B) (Figure 43). In method B, the arylidene compounds **175a**–**f** were formed at first and reacted under reflux in acetic acid with **120a** in the presence of NH_4_OAc. It seems that 4-hydroxycoumarin (**120a**) reacted with NH_4_OAc to give 4-aminocoumarin, which then added to **175a**–**f** in a Michael addition reaction type, followed by cyclization via dehydration to give the 1,4-dihydro pyridine derivatives **176a**–**f**.

In 2009 Tu et al., prepared naphtho [2,3-*f*]quinoline derivatives by a one-pot three-component reaction of 2-aminoanthracene (**177**), aromatic aldehydes and 1,3-diketones under microwave irradiation in order to test their luminescent properties [129]. The derivative 4-Hydroxycoumarin (**120a**) resulted in the fused coumarin derivatives **178a**–**f**. These compounds exhibited good luminescent properties (Figure 44). In the mechanism proposed [130], a condensation of **177** with aldehyde gave imine intermediate **A**, which underwent addition of 4-hydroxycoumarin (**120a**) to the adduct **B**, The latter after tautomerization to **C**, and removing of amine **177,** led to α,β-unsaturated intermediate **D**. Addition of amine **177** to **D** and cyclization of the tautomer **F** of **E** resulted in adduct **G**. Water was eliminated from the latter and the final product **178** was obtained. Compounds **178a**–**f** exhibited good luminescent properties in ethanol solution and could be used as organic electroluminescent media.

In 2011, Shafiee and coworkers prepared 4-aminocoumarin (**13**) by melting of 4-hydroxycoumarin (**120a**) in the presence of ammonium acetate. Then, they synthesized the chromeno [4,3-*b*]quinoline derivatives **180a**–**m** by the reaction of **13** with 2-arylidenecyclohexano1,3-dione derivatives **179a**–**m** under heating at high temperature without solvent [131]. The Michael addition of 4-aminocoumarin (**13**) to α,β-unsaturated compound **179** gave the intermediate **A**, according to the proposed mechanism (Figure 45). Isomerization of **A** to **B**, followed by intramolecular cycloaddition to **C** and subsequent elimination of water resulted in the final 1,4-dihydropyridocoumarin derivatives **180**. The synthesized compounds were tested for their cytotoxic activity in human cancer cell lines (Hela, K562, LS180 and MCF-7). Some of them showed moderate cytotoxic capacity and, in parallel, very low calcium channel antagonist activity. Compound **180a** presented the highest antitumorial activity (IC_50_ = 25.4–58.6 μM).

In 2013, Su and coworkers prepared the dihydrochromeno [4,3-*b*]quinoline derivatives **182a**–**t** via the three-component reaction of 4-hydroxycoumarin (**120a**) with aldehydes and anilines **181a**–**g** catalyzed by the ionic liquid *L*-2-(2-hydroxymethyl)-1-(4-sulfobutyl)pyrrolidinium hydrogen sulfate ([HYSBPI].H_2_SO_4_ in water under microwave irradiation [132]. In the proposed mechanism, according to reference [130], a condensation of aniline (**181a**) with benzaldehyde (**51a**) gave the imine **A**. The intermediate **B** was formed by the addition of **120a** to **A**, followed by the removal of aniline to give the benzylidene intermediate **C**. Addition of aniline to **C**, followed by intermolecular cyclization, led to the cyclized adduct **E**. Elimination of water from the latter resulted in the final product **182a** (Figure 46). Oxidation of **182a** with DDQ afforded the pyridocoumarin **183a**. The antitumor activity of the prepared compounds was evaluated in human cancer cell lines (A-549 and MCF-7). They exhibited moderate antitumor activities with IC_50_ = 0.05–100 μmol/L.

Next year, Choudhury and coworkers prepared similar dihydrochromeno [4,3-*b*]quinoline derivatives by the multi-component reaction of 4-hydroxycoumarin (**120a**) with aldehydes and anilines in water catalyzed by Bi(OTf)_3_ (10 mol%) under microwave irradiation [133]. For the mechanism, they proposed the 1,2-addition of aniline to the alkylidene intermediate **C**, for the formation of imine **D**, followed by 6 π electrocyclization to the intermediate **E**. Tautomerization of the latter led to the dihydropyridine derivative **182a** (Figure 47). When some of the above reactions were performed without solvent by conventional heating at 140 °C, for 2–4 h the chromeno [4,3-*b*]quinoline-6-ones **183** were received, possibly by a radical mechanism. Treatment of some of the dihydropyridocoumarin derivatives with NBS at room temperature resulted rapidly in a more clean reaction to the chromeno [4,3-*b*]quinoline-6-ones **183**. The fluorescent properties of the synthesized compounds were studied in different solvents. It was found that derivatives **182**, **184** are more fluorescent than the corresponding **183** analogs.

For the transformation of the above referred solvent-free oxidation of compounds **182**, **184** to the chromeno [4,3-*b*]quinoline-6-ones **183,** the authors suggested that a radical mechanism is taking place with the parallel reduction of Bi(III) to Bi(0), as depicted in Figure 48.

The same year, Pal and coworkers utilized Fe_3_O_4_@SiO_2_ nanoparticles as catalyst for the synthesis of 1,4-dihydropyridocoumarins **176g**–**m** from the one-pot multi-component reaction of 4-hydroxycoumarin (**120a**), aldehydes and NH_4_OAc on water [134]. The catalyst could be removed with a magnet and reused further. A plausible mechanism is depicted in Figure 46. By the binding of Fe_3_O_4_@SiO_2_ nanoparticles to the carbonyl oxygen, the carbonyl activity increased, facilitating the nucleophilic attack to the aldehyde and the formation of the intermediate Knoevenagel alkylidene product **175** (Figure 49). This adduct underwent a Michael addition from another molecule of **120a** to give intermediate **C**. Then ammonia reacted with **C** and amino-intermediate **D** was obtained, which under cyclization and elimination of water resulted in the desired product **176**. 

In 2015, Sashidhara et al., demonstrated a one-pot procedure for the synthesis of chromeno [4,3-*b*]quinoline-6-ones **183**, **185** under microwave irradiation in the presence of molecular iodine (10 mol%) via a three-component reaction of 4-hydroxycoumarin (**120a**), aromatic aldehydes and anilines [135]. Condensation of aldehydes with aniline led to the Schiff base **A**. Nucleophilic attack of 4-hydroxycoumarin on imine **A** gave unstable adduct **B**, which then underwent rearrangement to afford the intermediate **E** via the transition states **C** and **D** (Figure 50). A 1,3-H shift resulted in the 1,4-dihydropyridocoumarin **F**, which upon oxidation in the presence of iodine furnished the final product **183**.

In 2016, Yin and coworkers presented the three component synthesis of dihydrochromeno [4,3-*b*]pyrazolo [4,3-*e*]pyridines **188a**–**n** or chromeno [4,3-*b*]pyrazolo[4,3-*e*]pyridines **189a**–**g** from 4-hydroxycoumarin (**120a**), aldehydes and 5-amino-3-methyl-*N*-phenylpyrazole (**187a**)/5-amino-3-methylisoxazole (**187b**), depending on the conditions, refluxing in AcOH-MeCN or heating at 140 °C in AcOH-DMSO [136]. According to the proposed mechanism, a Knoevenagel reaction of 4-hydroxycoumarin (**120a**) with benzaldehyde (**51a**) gave the benzylidene-adduct **A**, which reacted **187a** in a Michael reaction furnishing intermediate **B** (Figure 51). Intramolecular cyclization by the addition of amine group to the coumarin 4-carbonyl and elimination of water resulted in the intermediate **C**. Tautomerization of the latter gave dihydro derivative **188a**, which upon oxidation led to the pyridocoumarin product **189a**.

The same year, Khurana and coworkers reported an analogous synthesis of dihydrochromeno [3,4-*e*]isoxazolo [5,4-*b*]pyridine-6-ones **188n**, **190a**–**n** in excellent yields via a one-pot three-component reaction of 4-hydroxycoumarin (**120a**), aldehydes and 5-amino-3-methylisoxazole (**187b**) in the ionic liquid 1-butyl-3-methylimidazolium hydrogen sulfate ([C4mim] [HSO_4_]) under heating at 80 °C or ultrasonic irradiation at room temperature [137]. For the mechanism, a different approach has been proposed, in comparison to the above reaction. The iminium intermediate **A** was formed by the condensation of 5-amino-3-methylisoxazole (**187b**) with benzaldehyde (**51a**) in the presence of acidic ionic liquid. Nucleophilic addition of **120a** to **A** gave the unstable transition state **B**, which underwent cleavage furnishing the benzylidene intermediate **C** (Figure 52). A 1,3-Addition of 5-amino-3-methylisoxazole (**187b**) to **C** resulted in the intermediate **D**, which under intramolecular cyclization led to the intermediate **E**. By the 1,3-H shift of the latter the final product **188n** was obtained. The authors confirmed the above mechanism performing the reaction of preformed alkylidene compound **A** with **187b**, which resulted in the expected final product **188n**.

Ionic liquid, functionalized by silica@γ-Fe_2_O_3_ magnetic nanoparticles (MNP), was used also by Mahdavi and coworkers as a catalyst for the synthesis of 6*H*-chromeno [4,3-*b*]quinoline-6-ones **182**, **184** via multi-component reaction of 4-hydroxycoumarin (**120a**), aromatic aldehydes and aromatic amines (Figure 53). The catalyst (IL-SiO_2_@MNP) was prepared by the reaction of 1-butyl-3-(3-(trimethoxysilyl)propyl-1*H*-imidazol-3-ium chloride with iron oxide nanoparticles coated by silica. The nanocatalyst could be separated after the completion of the reaction by a magnet and showed activity up to 10 times [138].

The same year, Foroumadi and coworkers synthesized the coumarin-fused dihydropyridinones **192a**–**i** via a multi-component reaction of 4-hydroxycoumarin (**120a**), ammonia, aromatic aldehydes, and Meldrum’s acid (**90e**) in refluxing propan-1-ol [139]. In the proposed mechanism, the benzylidene intermediate **A** was formed by the Knoevenagel reaction of Meldrum’s acid (**90e**) with aldehyde **51a** (Figure 54). The amination reaction of 4-hydroxycoumarin with ammonia generated the 4-aminocoumarin (**13**), which by a Michael reaction to **A** led to the intermediate **B**. Intramolecular cyclization of the latter under loss of acetone and CO_2_ resulted in the final product **192a**.

In 2018, Sayahi et al., demonstrated the same multi-component reaction for the synthesis of coumarin-fused dihydropyridinones **192a**–**h** in the presence of SBA-15-SO_3_H, a mesoporous material with nanochannels, as catalyst [140]. This reaction completed in less reaction time with better yield than in the former work (Figure 55).

In 2019, Mohammadpoor-Baltork and coworkers demonstrated the synthesis of chromeno [4,3-*b*]quinoline-6-one derivatives **183**, **185** via a one-pot three-component reaction of 4-hydroxycoumarin (**120a**) with aldehydes and anilines catalyzed by halloysite nanoclay under solvent-free conditions [141]. Except for the products presented in Figure 56, they also prepared analogous derivatives using two different anilines with aldehydes, or two different aldehydes with aniline or aromatic diamines with aldehydes or dialdehydes with anilines. In the proposed mechanism, the activated benzaldehyde **A** condensed with aniline to give imine **B**. The nucleophilic attack of **120a** afforded the intermediate **C**, which upon elimination of aniline resulted in the benzylidene intermediate **D**. Recondensation with aniline led to **E**. Electrocyclisation afforded **F,** and by an 1,3-H shift, dihydro-adduct **G** was formed. Anomeric-based oxidation in the presence of the catalyst, through **H**, resulted in the final product **185i**.

The same year, Zeynizadeh and Rahmani reported the Hantzsch synthesis of 1,4-dihydropyridocoumarin derivatives **176,** via the multi-component reaction of 4-hydroxycoumarin, aromatic aldehydes and ammonia, in the presence of a clay magnetic nanocatalyst [(NiFe_2_O_4_@Cu)SO_2_(MMT)] resulted from the reaction of sulfonated montmorillonite SO_2_(MMT) with copper immobilized nickel ferrite (NiFe_2_O_4_@Cu) [142]. The activated with clay nanocatalyst benzaldehyde **A** reacted in a Knoevenagel reaction with **120a** to the benzylidene adduct **B**, which reacted with ammonia to give the imine **C**. This by activation with clay reacted with a second molecule of **120a** and furnished the Michael adduct **D**. Tautomerization of the latter led to the enamine **E**, which under cyclization resulted in the final product **176a** (Figure 57).

The same research group reported, in 2019, the use of another magnetic nanocatalyst, the sulfonated Ni-nanocatalyst NiFe_2_O_4_SiO_2_@SO_3_H, for the Hantsch synthesis of the same 1,4-dihydropyridocoumarins **176** (Figure 58) [143]. A similar mechanism such as Figure 53 was proposed under activation with the sulfonated Ni-nanocatalyst.

Recently, Ghosh and coworkers utilized graphene oxide (GO) as a catalyst for the one-pot three component synthesis of chromeno [4,3-*b*]quinoline-6-one derivatives **183**, **185** from 4-hydroxycoumarin (**120a**), aldehydes and anilines under solvent-free conditions [144]. GO could be recovered and reused up to five runs without losing the catalytic activity. In the proposed mechanism, the condensation of the activated benzaldehyde **A** with 4-hydroxycoumarin (**120a**) furnished after dehydration the unstable adduct **C**, which underwent nucleophilic attack from p-toluidine (**181b**) to give intermediate **D** (Figure 59). After cyclization to **E** and dehydration, the intermediate **F** oxidized in the presence of GO and finally resulted in 9-methyl-7-phenyl-6*H*-chromeno [4,3-b]quinoline-6-one (**183o**).

The same year, Lee and coworkers reported the synthesis, between other fused pyridine derivatives, of pyrido [3,2-*c*]coumarin derivatives **194a**–**c** via the one-pot multi-component reaction of 4-hydroxycoumarin (**120a**), *N*,*N*-dimethylformamide dimethyl acetal, dimedone and ammonium acetate catalyzed by In(OTf)_3_ under solvent-free conditions [145]. The derivatives were tested for their photophysical properties as photoluminescent probes. 4-Hydroxycoumarin (**120a**) attacked the iminium ion **B** generated from **193** under In(OTf)_3_ catalysis (Figure 60). Elimination of methanol from the intermediate **C** formed led to the intermediate **D**, which was isolated by the authors in the control experiment. Nucleophilic addition of the enol **90g′** gave Michael’s reaction adduct **E**. The latter reacted with ammonia, generated from ammonium acetate, to afford intermediate **F**. Intramolecular condensation furnished the dihydro-intermediate **G**. Elimination of dimethylamine resulted in the final product **194a**.

In 2003, Guillaumet and coworkers synthesized pyrido [2,3-*c*] and pyrido [3,2-c]coumarins by the reactions of 3-hydroxycoumarins **195a**–**e** or 4-hydroxycoumarin (**120a**) with β-aminoketones **196a**–**f**, having the carbonyl group protected [146]. Condensation of **195a** with the amine **196a** in the presence of catalytic amount of camphorsulfonic acis (CSA) in toluene under reflux and Dean–Stark apparatus gave the intermediate **A** (Figure 61). Cyclization in the presence of BF_3_ etherate under reflux led to intermediate **B**, which under disproportionation resulted in a mixture of **197a**/**198a** (56:44). The similar reactions of **120a** with **196a**,**b**,**f** afforded the pyrido [3,2-*c*]coumarins **199a**–**c**. When **198a** was treated with DDQ, this furnished pyrido [2,3-c]coumarin (**197a**). When the reactions of **195a** with protected aminoketones, **196a**,**b**,**d**, were treated in the third step with NaBH_3_CN, the tetrahydopyrido [2,3-*c*]coumarins **198a**–**c** were isolated. 

##### Synthesis with Krohnke’s-Type Reaction

Krohnke’s reaction is the reaction of α-pyridinium methyl ketone salts with α,β-unsaturated ketones in the presence of ammonium acetate in acetic acid for the synthesis of substituted pyridines [147,148,149]. 1,3-Dicarbonyl compounds are used also in place of pyridinium salts for the synthesis of pyridines under these reactions [149].

Application of Krohnke’s reaction in the case of 4-hydroxycoumarins **120** as a one-pot reaction with chalcones **200** and ammonium acetate resulted in the fused pyrido [3,2-*c*]coumarins **201** [150]. In this reaction, the enamine **13**, in situ formed from **120a** and ammonium acetate, added in a Michael reaction to the chalcone **200a** and gave the intermediate **A** (Figure 62). Tautomerization of the latter, dehydration and aromatization furnished the final product **201a**.

In 2010, Dawane, Konda and coworkers reported an analogous Krohnke’s reaction of 4-hydroxy-7-methylcoumarin (**120j**) with chalcones **202**, containing a pyrazole moiety, in poly(ethylene glycol) (PEG-400) in the presence of ammonium acetate for the synthesis of the pyrido [3,2-*c*]coumarins **203** (Figure 63) [45]. The proposed mechanism for this reaction was the same as that presented in Figure 62. The synthesized derivatives were evaluated for their antimicrobial properties. The antibacterial activity was checked against bacteria *Escherichia coli*, *Salmonella typhi*, *Staphylococcus aureus* and *Bacillus subtilis*. *Salmonella typhimurium*, Antifungal activity was tested against *Aspergillus niger*, *Aspergillus flavus*, *Penicillium chrysogenum*, and *Fusarium moniliforme*. Compounds **203b**, **203d**–**f**, **203h**, **203j**, **203l** showed good antibacterial activity against one or more bacteria. Most of the compounds presented inhibitory effect against fungi.

In 2011, Brahmbhadtt and coworkers applied Krohnke’s reaction, changing the chalcone to 2-arylidene tetralones **204a**–**c**, and synthesized the fused aza-phenanthrocoumarins **205a**–**l** [151]. According to the proposed mechanism, the anion **A** formed from 4-hydroxycoumarin (**120a**) and ammonium acetate reacted with **204a** in a Michael addition to give intermediate **B** (Figure 64). Addition of ammonia furnished the adduct **C**, which cyclized to **D**, through the addition of amine-group to the coumarin carbonyl. Dehydration led to the 1,4-dihydro intermediate **E**. Oxidation of the latter resulted in the final product **205a**. Recently, the same group demonstrated the crystal structure of compound **205a** [152]. All the compounds were tested for their antibacterial activity against *Escherichia coli* (gram − ve bacteria) and *Bacillus subtilis* (gram + ve bacteria) and antifungal activity against *Candida albicans* (Fungi). Compounds **205i**–**l**, with chlorine atom in coumarin moiety, showed better activity against *E. coli* and *B. subtilis*. All the compounds presented moderate activity against fungi *C. albicans*, except compound **205e** with poor activity and **205b** with no activity.

In 2014, Yin and coworkers demonstrated the one-pot synthesis of pentacycle coumarin derivatives **207a**–**k** from the multi-component reaction of 4-hydroxycoumarin (**120a**), 2-hydroxychalcones **206a**–**k** and aqueous ammonia in refluxing *n*-propanol under catalyst-free conditions [153]. In the proposed mechanism, the intermediate **A** was formed by a Michael reaction of **120a** to the chalcone **206a** (Figure 65). Amination with ammonia furnished the intermediate **B**, which through the tautomer **C** cyclized to dihydropyridocoumarin **D** under elimination of water. Intramolecular addition of a hydroxyl group to the imino moiety resulted in the final pentacycle product **207a**.

In 2019, Giri and Brahmbhadtt synthesized bipyridyl-fused coumarins **209a**–**l** by the Krohnke’ reaction of 4-hydroxycoumarins **120** with chalcones **208a**–**c** and ammonium acetate in glacial acetic acid [154]. In the proposed plausible mechanism, the intermediate **B** was formed by the Michael reaction of carbanion **A** to the chalcone **208a**. Addition of ammonia to the 4-carbonyl of coumarin (Figure 66), and not to the benzoyl carbonyl as in Figure 62, gave the intermediate **C**, which cyclized to **D** upon nucleophilic addition of amine to the benzoyl carbonyl. Dehydration of the latter gave 1,4-dihydropyridocoumarin **E**, which by oxidation resulted in the final product **209a**. The synthesized compounds were tested for their antimicrobial activity against gram-positive bacteria (*Bacillus subtilis* and *Staphylococcus aureus*) and gram-negative bacteria (*Escherichia coli* and *Salmonella typhimurium*) and antifungal activity against *Aspergillus niger* and *Candida albicans*. Compounds **209c**, **209f**, **209i** exhibited the better antimicrobial activity.

#### 2.1.3. Synthesis from Various Coumarin Derivatives

In 1994, Heber and Berghaus reported the synthesis of pyridocoumarins **212a**–**c** by the treatment of 4-aminocoumarin derivatives **210a**–**c** with a mixture of DMF and phosphorus oxychloride under Vilsmeier conditions [74]. Reduction of **212b**,**c** with sodium cyanoborohydride resulted in azacannabinoids **213b**,**c** (Figure 67).

In 2003, Al-Omran et al., synthesized coumarin derivative **215** by the reaction of 3-cyano-4-methylcoumarin (**214**) with dimethylformamide dimethylacetal (DMFDMA) under reflux in xylene [155]. Coumarin derivative **215** reacted with benzotriazole-1-ylacetic acid hydrazide under fusion to give the triazolo-fused pyridocoumarin **217** through the intermediates **A** and **B** (Figure 68). The reaction of **215** with hydrazine hydrate resulted in *N*-aminopyridine derivative **216**. Treatment of **216** with chloroacetylacetone led to pyridotriazine derivative **218**. The reaction of **215** with 2-aminopyridine, urea, glycine, 2-aminocrotonitrile or cyanothioacetamide resulted in the pyridocoumarin derivatives **219**–**223**. The compounds were tested for their antifungal activity against *Aspergillus niger* and for their antibacterial activity against *Escherichia coli*, *Staphylococcus aureus* and *Bacillus subtilis*. Most of the products showed antibacterial and fungicidal activities.

In 2005, Ismail and Noaman synthesized the chromeno [3,4-*c*]pyridone **226** by the treatment of 3-carboxamidocoumarin **225** with malonitrile (**91**) in the presence of ammonium acetate in refluxing ethanol [156]. Coumarin derivative **225** was obtained by the Perkin reaction of 2-ethyl cyanoacetanilide (**224**) with salicylaldehyde (**51w**) in acetic anhydride and sodium acetate under reflux (Figure 69). The compound **226** was tested for antifibrotic activity, but it showed high fibrotic potential.

The same year, Sun and coworkers isolated, in a three-step procedure, the angularly-fused benzofuropyridinocoumarins **230** in 70–80% yield by refluxing of 2′-cyano-4-phenoxycoumarins **229** with excess of orthoesters [157]. The reaction proceeded through the imine **A** (Figure 70). The products **230a**–**g** were tested for their anti-inflammatory, analgesic and anti-microbial activities. Compounds **230b** and **230f** showed significant inhibition of inflammation (78–97%), whereas compounds **230a**, **230b**, **230f**, **230g** presented interesting analgesic activity (one-third of the protection caused by acetylsalicylic acid). Compound **230f** was the most promising.

In 2008, Glasnov and Ivanov synthesized dimethyl and diethyl 5-oxo-1,2-dihydro-5*H*-chromeno [4,3-*b*]pyridine-2,3-carboxylates **233a**–**n** by the reaction of 4-amino-3-formylcoumarins **231a**–**g** with acetylenedicarboxylates **232** and **83** in the presence of triphenylphosphine [158]. The final products were formed by an intramolecular Wittig reaction of the intermediate **A** (Figure 71).

The same year, Majumdar et al., reported the synthesis of 6a,7,8,12b-tetrahydro-6*H*-chromeno [3,4-*c*]quinolin-6-ones **237a**–**f** from 3-(2-bromoanilinomethyl)coumarins **236a**–**f** [159]. The latter were prepared by the reaction of 2-bromoanilines **235a**–**f** with 3-chloromethylcoumarin (**234**). The formation of products **237** could be explained by the generation of aryl radical **A**, which by a 6-*endo* trig cyclization produce the radical **B**, stabilized by the adjacent carbonyl (Figure 72). Protonation of **B** led to the final compound **237**.

In 2010, Kulkarni and coworkers synthesized the coumarin analogues of protoberberine alkaloids **245a**–**f** by a Mannich reaction of 1,2,3,4-tetrahydroisoquinoline derivatives of coumarin **244a**–**f** [160]. Compounds **244** were obtained by the reduction of hydrochloric acid salts **243** with NaBH_4_. A Bischler–Napieralski reaction of amides **241** had furnished dihydroisoquinoline derivatives **242** (Figure 73). Compounds **244** and **245** were tested for their antibacterial activity. Compounds **244e**,**f** and **245e**,**f** presented selectivity towards gram-positive bacteria *Staphylococcus aureus* and *Aspergillus niger*.

In 2012, Ghorab and coworkers reported the synthesis of benzo-fused chromeno [3,4-*c*]pyridone **248** by the treatment of 3-carboxamidocoumarin derivative **247** with malonitrile (**91**) [161] following the above presented similar work [99]. A Perkin reaction of 3-ethyl cyanoacetanilide (**246**) with 2-hydroxy-1-naphthaldehyde furnished the coumarin derivative **247** (Figure 74). Compound **248** presented low anticancer activity (IC_50_ = 245.7 μM) against Ehrlich Ascites carcinoma (EAC).

In 2017, Shi and coworkers utilized silica sulfuric acid (SSA) as a catalyst to synthesize the coumarin derivatives **250a**–**g** with hetero [5]helicene-like conformation [162]. This synthesis was achieved by the reaction of 3-ketobenzo[f]coumarin derivatives **249a**–**e** with aminopyrazoles **187** in the presence of DDQ in DMF under microwave irradiation (Figure 75).

In 2018, Borah synthesized the fused 1,2,3,4-tetrahydropyridocoumarin derivatives **255a**–**d** using the hetero Diels–Alder strategy [163]. The hetero Diels–Alder reaction of the Knoevenagel condensation adducts **254a**–**d** under reflux resulted in the final products **255a**–**d**. Compounds **254a**–**d** were achieved from the reaction of 3-allylamino-3-formyl coumarins **253a**,**b** with *N*,*N*-dimethylbarbituric acid (**90a**) and barbituric acid (**90b**) in the presence of piperidine as a base (Figure 76). The 3-allylamino-3-formylcoumarins **253a**,**b** were prepared from the reaction of 4-chloro-3-formylcoumarin (**251a**) with allylamines **252a**,**b**.

In a Chinese patent of 2019 [164], Li, Yang and Chen referred the synthesis of pyrido [3,4-*c*]coumarins **258a**–**v** by the Diels–Alder reaction of 3-acetoxyiminocoumarins **256a**–**o** with intermediate alkynes **257a**–**e** in the presence of dichloro(pentamethylcyclopentadienyl)rhodium(III) dimer as a catalyst (2.5 mol%) (Figure 77).

In 2020, Vala and coworkers performed the reaction of 3-ethylaminomethyl-4-hydroxycoumarins **259a**–**d** with aroylmethyl pyridinium salts **261a**–**d** in the presence of ammonium acetate and acetic acid and synthesized the 2-arylpyrido [3,2-*c*]coumarins **262a**–**p** [165]. Compounds **259** were prepared by a Mannich reaction of 4-hydroxycoumarins **120** with formaldehyde and ethylamine, while the pyridinium salts **261** were obtained by treating phenacyl bromides **260** with pyridine (Figure 78).

In 2013, Brahmbhatt and coworkers utilized, also, the 3-ethylaminomethyl-4-hydroxycoumarins **259a**–**d** to synthesize in moderate yields 2-(2-oxo-2H-chromen-3-yl)-5H-chromeno [4,3-b]pyridin-5-ones **264a**–**l** through the reaction with the pyridinium salts **263a**–**c**, in the presence of ammonium acetate and acetic acid [43]. For the reaction pathway, decomposition of **259a** resulted in the intermediate coumarin methide **A**, which then reacted with **263a** in the presence of NH_4_OAc and AcOH to give the 1,5-dicarbonyl intermediate **B**. The latter was converted to the final product **264a** via a Krohnke’s-type reaction (Figure 79). The new compounds **264a**–**l** were tested for their antibacterial activity and presented potent inhibitory activity against gram-positive bacteria, *Bacillus subtilis* and *Staphylococcus aureus*. They showed also appreciable activity against gram-negative bacteria, *Escherichia coli* and *Salmonella typhimurium*, as well as antifungal activity against *Aspergillus niger* and *Candida albicans*. Compounds **264e**, **264f**, **264i**, **264k**, **264l** were found to be the more proficient.

In the same presentation, Brahmbhatt and coworkers used another route for the synthesis of compounds **264a**–**l** with the 4-chloro-3-formylcoumarins **251a**–**d** and pyridinium salts **263a**–**c** as starting materials [43]. The reaction of **251a** and **263a** resulted in the intermediate **C**, which then was converted to the final product **264a** (Figure 79).

The 4-chloro-3-formylcoumarins **251**, used in the above reactions, are very versatile tools for the synthesis of fused pyridocoumarins [67]. In 1995, Heber et al., synthesized 3-substituted [1]benzopyrano [4,3-b]pyridine-5-ones **28j**–**l** from the treatment of 4-alkylamino-3-alkenylcoumarins **268a**–**k** under Vilsmeier conditions [166]. This reaction proceeded probably via the intermediate dimethyliminium salt **269**, followed by a nucleophilic attack of chloride anion to the *N*-alkyl moiety with subsequent electrocyclization and elimination of alkyl chloride and dimethylamine (Figure 80). Compounds **268a**–**k** were obtained by the conjugated addition–elimination reaction of 3-alkenyl-4-chlorocoumarins **266a**–**d** with alkylamines **267a**–**h**. The coumarin derivatives **266a**–**d** were prepared by the Wittig reaction of **251a** with the ylides **265a**–**d**.

In 2006, Wu et al., reported that the reaction of 4-chloro-3-formylcoumarin **251** with aryl isocyanides **270a**–**c** led to the synthesis of chromeno [4,3-*b*]quinolin-6-ones **170**, **172** [167]. A possible mechanistic pathway was proposed according to Figure 81. The amine **B** was formed from the addition of methanol to isocyanide **270**. 1,4-Addition of **B** to **251,** followed by elimination of the chloride anion, gave the intermediate **C** (Figure 81). After cyclization through attack to aldehyde moiety the intermediate **D** was formed, which led to the alcohol derivative **E**. Elimination of water resulted in the products **170**, **172**. In the case of 4-diethylaminophenyl isocyanide (**270c**) the dihydro derivatives **271a**,**b** were isolated possibly via 1,3-H shift from the intermediate **E**.

In 2011, Gopikrishna and coworkers reported the synthesis of fused chromeno [4,3-*b*]quinolin-6-ones **170**, **172** by the one-pot reaction of anilines **181** with 4-chloro-3-formylcoumarin (**251a**) [168]. In the plausible proposed mechanism, the nucleophilic addition of aniline (**181a**), followed by elimination of chlorine, furnished *N*-aryl intermediate **A** (Figure 82). Activation of the formyl group with the in-situ generated HCl gave through the **B** the cyclization intermediate **C**. Elimination of water from the isomerization intermediate **D** led to the final product **170a**.

One year later, Pal and coworkers performed an analogous reaction preparing the 6*H*-1-benzopyrano[[4,3-*b*]quinolin-6-ones **170**, **172** by sonication in methanol (Figure 83). Compounds **170**, **172** were tested for their antiproliferative properties against human chronic myeloid leukemia cells, human colon carcinoma cells, breast cancer cells and human neuroblastoma cells, and some of them were found to be active [48].

The same year, Bhuyan and coworkers prepared the tetrazole-fused pyrido [3,2-*c*]coumarin derivatives **273a**–**j** by the one-pot three-component reaction via the intramolecular 1,3-cycloaddition reaction of azides to nitriles [169]. The intermediate azides **B** was obtained by the reaction of sodium azide with 3-alkenyl-4-chlorocoumarin **A** (Figure 84). The reaction of 4-chloro-3-formylcoumarin (**251a**) with the nitriles **91**, **271a**–**i** led to the intermediates **A**.

In 2018, El-Agrody and coworkers synthesized derivatives containing indole fused with coumarin moiety such as **275** and **276** (Figure 85). The reaction of **251a** with 5-aminoindole (**274**) in the presence of Et_3_N under heating resulted in the synthesis of chromeno [4,3-*b*]pyrrolo [3,2-*f*]quinoline-12(3*H*)-one (**275**). The three-component reaction of **274** with 4-hydroxycoumarin (**120a**) and m-nitrobenzaldehyde (**78k**) in the presence of *N*-chlorosuccinimide under microwave irradiation led to 13-(3-nitrophenyl)-6,13-dihydrochromeno [4,3-b]pyrrolo [3,2-f]quinoline-12(3*H*)-one (**276**). The synthesized compounds were screened for their cytotoxic activity against the human cervix carcinoma cell line (KB-3-1) [170]. Compound **276** presented higher potent activity (IC_50_ = 7.9 μM) in comparison to the positive control compound (+)-Griseofulvin (IC_50_ = 19.2 μM).

The same year, Kolita and Bhuyan demonstrated the synthesis of pyrido [3,2-*c*]coumarins **262**, or **92k**, **277a**,**b** or **279a**–**c** from the reaction of 4-chloro-3-formylcoumarin (**251a**) with aryl methyl ketones **87** or ethyl acetoacetate (**90k**) or ethyl cyanoacetate (**272b**), respectively, in the presence of NH_4_OAc under microwave irradiation and solvent-free conditions [171]. An aldol condensation between **251a** and **87a** possibly occurred to give the intermediate **A**. The latter in the presence of NH_4_OAc furnished the 4-aminocoumarin intermediate **B** (path A) or the imine intermediate **C** (path B). Intramolecular condensation of **B** or intramolecular *N*-substitution of C led to the pyrido [3,2-*c*]coumarin **262a** (Figure 86). In the case of ethyl acetoacetate (**90k**), a Knoevenagel condensation to intermediate **D** followed by a condensation of amine group with carbonyl resulted in the final products **92k** or **277a**,**b**. When ethyl cyanoacetate (**272b**) was used, a condensation intermediate **E** again might have been formed. Addition of amide to cyanide group of **E** afforded via cyclization the products **278a**–**c**. There was evidence from the H^1^-NMR for the imine tautomeric form of those compounds and not for the amine tautomers **279a**–**c**.

In 2011, Iarosenko, Langer and coworkers used an analogue of **251a**, the 3-acyl-4-chlorocoumarins **280a**,**b**, for the synthesis of 3,4-fused pyridocoumarins **283a**–**c**, **285a**,**b**, **287a**–**l**, **289**, **290** by cyclocondensation with electron-rich aminoheterocycles [172]. The reactions of amines **281a**,**b**, **284a**,**b**, **286a**–**g**, **288** resulted in fused pyrido [2,3-*c*]coumarin derivatives via an attack of the internal enamine β-carbon of **A** for the formation of intermediate **B** (Figure 87). Intramolecular attack of amine to the carbonyl of **B** led, under cyclization, to **C** and elimination of water to the final products. In the case of 5-amino-3-methyl-1-phenylpyrazole (**187a**), the regio-isomer of the fused pyrido [3,2-*c*]coumarin derivative **290** was synthesized probably due to the more aromatic character of amine.

Earlier in 2004, Trimarco and coworkers utilized another 3-formylcoumarin derivative, the 4-azido-3-formylcoumarin (**291**), for the synthesis of fused pyrido [3,2-*c*]coumarin derivatives **294a**–**f** [173]. The reaction of **291** with the enamines **292a**–**f** resulted in the amidines **293a**–**f** via the 1,3-cycloaddition intermediate **A** (Figure 88). The amidines, by treating with catalytic amount of MeONa in refluxing MeOH, led to the final products **294a**–**f**. The intermediates **295e**,**f** (30%) were isolated in 2 h in the case of **293e**,**f**, along with the hydrolysis adduct, 4-amino-3-formylcoumarin (**231a**) (35%) and then refluxed for an additional time of 2 h to give **294e**,**f**.

### 2.2. Pyranone Ring Formation

The formation of a pyranone ring could be obtained using the cyclization of suitable aryl-substituted pyridine or piperidine derivatives. Phenol derivatives, also, as well as salicylaldehydes could be the starting material, resulting in the construction of the pyranone ring.

#### 2.2.1. Synthesis from Pyridine or Piperidine Derivatives

Khan et al., isolated the pyrido [3,4-*c*]coumarin **297** via the cyclization of the chloride of 4-(*o*-methoxyphenyl)lutidine-3-carboxylic acid sulfate **296** with aluminum chloride in nitrobenzene at 50 °C for 2 h [174], in an attempt to prepare 2-azafluorenone derivatives. The intermediate chloride was prepared by refluxing **296** in thionyl chloride for 10 min (Figure 89).

An analogous starting material, the 3,5-dicyano-4-(*o*-methoxyphenyl) pyridines **298a**–**c**, were used by Courts and Petrow for the synthesis of pyrido [3,4-*c*]coumarin **299a**–**c** [175]. The cyclization was performed by refluxing in hydrobromic acid for 2 h via the intermediate imine **A** (Figure 90).

Gorlinger and coworkers, in 2006, utilized the pyridine derivative **300** to synthesize the pyrido [3,4-*c*]coumarin **302** by the reaction with novaldiamine (**301**) (Figure 91). These compounds together with other prepared were tested for in vitro antimalarial activity against Plasmodium falciparum strain Dd2 and 3D7 [50]. Compounds **302** and **303** presented quite good activity with IC_50_ = 1.1 μM, 3.4 μM and 6.2 μM, 7.0 μM, respectively.

In 2007, Kelly and coworkers synthesized the natural product santiagonamine (**315**) using the pyridine derivative **304** (prepared from pyridine-3-carboxylic acid) as starting material, and benzaldehyde derivatives **305** or **307** (prepared from isovanillin) via a Pd-catalyzed Ullmann cross-coupling reaction [57]. After deprotection of **306**, Wittig reaction of **308** to **309**, bromination to **310**, cyclization in the presence of TFA to the pyrido [2,3-*c*]coumarin **311**, the **312** was obtained by photocyclization. Stille reaction of **312** with allyl tributyltin gave the allyl derivative **313**. Transformation of the latter with OsO_4_ and sodium periodate afforded aldehyde **314**, which under reductive amination with dimethylamine led to santiagonamine (**315**) (Figure 92). This was the first total synthesis of santiagonamine in 12 steps from isovanillin and 2.6% overall yield. 

In 1966, Pars et al., synthesized the nitrogen analogs of tetrahydrocannabinol **319** [176]. The Pechmann-type reaction of olivetol (**316**) with 4-carbethoxy-*N*-methyl-piperid-3-one hydrochloride (**317**) in the presence of concentrated sulfuric acid and phosphorus oxychloride resulted in the tetrahydropyrido [4,5-*c*]coumarin **318** (Figure 93). Treatment of the latter with MeMgI in anisole led to **319**.

Mandal et al., used the piperidin-4-one derivatives **320** for the synthesis of fused tetrahydropyrido [3,4-*c*]coumarin derivatives **322a**–**c** and **324a**–**d** in order to study their fungicidal activity against *Xanthomonas malvacearum*, *Fusarium maniliform*, *Rhizoctonia solanis*, *Powdery mildew* of cucumber, *Phytopthora* infection of tomatoes and grey mold of beans [177]. The Pechmann reaction of **320** with m-cresol (**321**) or a-naphthol (**323**) led to the fused coumarin derivatives **322a** and **324a**, respectively (Figure 94). Methylation of them with Me_2_SO_4_ or MeI resulted in the *N*-methyl derivatives **322b** or **324b**, while *N*-acylation with acetic anhydride or propionic anhydride gave *N*-acyl derivatives **322c** or **324c**,**d**, respectively. Tetrahydrobenzopyridicoumarins **324** presented higher fungicidal activity than compounds **322**. The substituents in the amine group led to a lowering of the fungicidal action in green plants.

The phenols **A** derived after the base-catalyzed rearrangement of 3-carbomethoxy *N*-(aryloxy) pyridinium tetrafluoroborate **327a**,**b** cyclized spontaneously to the fused pyrido [3,2-*c*]coumarins **124d**,**k,** according to Abramovitch and coworkers [178]. The *N*-(aryloxy) pyridinium salts **327a**,**b** were prepared by the reaction of pyridine-*N*-oxide **325** with the diazonium salts **326a**,**b** in dry acetonitrile (Figure 95).

In 2007, the fused pyrido [3,2-*h*]coumarin **330** and pyrido [3,2-f]coumarin **332** were prepared by our group from the reaction of triphenylphosphine (**329**) and DMAD (**232**) with quinolinol-8 (**328**) and quinolinol-6 (**331**), respectively, as starting material [179]. The coumarin skeleton is possibly produced by lactonization of the intermediate **D**, according to an analogous reaction of phenols by Yavari et al. [180]. Intermediate **D** was achieved by an 1,2-H shift and elimination of PPh_3_ from **C**, which was formed by the reaction of intermediates **A** and **B** (Figure 96).

#### 2.2.2. Synthesis from Phenol or Salicylaldehyde Derivatives

In 1998, El-Saghier et al. reported the synthesis of dihydro pyrido [3,4-*c*]coumarin derivatives **335a**–**d** starting from *o*-hydroxyarylidenemalononitrile **333a**,**c** or *o*-hydroxyarylidenecyanoester **333b**,**d** and ethyl 2-(4-aminosulfonylcarbanilide)acetate (**334**) in the presence of piperidine [181]. A transesterification of **334** with phenol followed by addition of the produced carbanion to the vinyl group is probably responsible for the formation of intermediate **A**. Addition of amine of benzanilide moiety to the cyano group led to the product **335a**–**d**, as referred in the reference (Figure 97). Possibly, these products are in the tautomeric form of **337a**–**d,** due to the proton peak of the dihydropyridine moiety at 4.50–5.10 ppm. The same products were synthesized, also, by the reaction of coumarin-3-(4-aminosulfonyl) carbanilide (**336a**) or benzo[f] coumarin-3-(4-aminosulfonyl) carbanilide (**336b**) with malononitrile (**91**) or ethyl cyanoacetate (**272b**). Analogous coumarin derivatives were obtained also in this work from the reaction of **336a**,**b** with some active methylene compounds.

Hosni et al., synthesized the fused pyrido [3,4-*c*]coumarins **339a**–**f** with thienyl or furyl substituents and studied their anti-inflammatory and analgesic activity [49]. The stirring of propenones **338**–**c** and malononitrile (**91**) in alcoholic potassium hydroxide at room temperature resulted in **338a**–**f** via the intermediate **A** (Figure 98). The compounds showed moderate potency in anti-inflammatory activity. They exhibited also analgesic activity more than the diclofenac, the standard reference. Compounds **339f** and **339e** were safer than diclofenac, having higher LD_50_.

Sviripa et al., prepared fused tetrahydropyrido [3,4-*c*]coumarins, such as **344**, **345**, 348, **349a**,**b**, **350a**–**c**, **351** via a Pictet–Spengler condensation of 4-(2-aminoethyl)coumarins, such as **343**, **347**, which have a C-7 activating amino or *N*,*N*-dialkylamino group [182]. 4-(2-Aminoethyl)coumarins were prepared from the corresponding phenols by treating with methyl sulfonic acid and methyl ester **341** (Figure 99). They studied, also, reactions of **343**, **347** with 5α-androstane-3-ones. Computational modelling has been used for the new molecules, analogs to 5α-dihydrotestosterone, as a tool to study 17-oxidoreductases for intracrine, androgen metabolism. The mechanism of the Pictet–Spengler reaction is interpreted for the case of cyclohexanone with **347** through the intermediates **A**, **B** and **C** for the synthesis of **349b** (Figure 99).

In 1969, Sakurai and Midorikawa utilized the condensation of salicylaldehydes with active methylene compounds for the synthesis of fused benzoquinazoline derivatives [183]. In the case of ethyl acetoacetate (**90k**) they have synthesized the fused pyrido [3,2-*c*]coumarin **353** from salicylaldehyde (**51w**) in the presence of ammonium acetate (Figure 100). Sakurai et al., also, synthesized the pyrido [3,4-*c*]coumarins **356a**–**z** and **357a**–**d** by the reaction of salicylaldehydes with ethyl cyanoacetate (**272b**) and aliphatic ketones **355** [184]. Salicylaldehydes and **272b** via the condensation intermediate **A** and the cyclization intermediate **B** resulted in 3-amidinocoumarin **354**, isolated during the heating for several min. When the heating was performed for 0.5–2 h, the reaction led to the final products **356a**–**z** and **357a**–**d**, through condensation with ketones, reaction of methylene group with C-4 of coumarin and subsequent dehydrogenation (Figure 100). 

The same group studied the reaction of salicylaldehydes with ethyl cyanoacetate (**272b**) in the presence of aldehydes **186** and ammonium acetate to obtain the amino-substituted pyrido [3,4-*c*]coumarins **358a**–**j** [185]. The reflux in ethanol was for 0.5–1.5 h. 3-Amidinocoumarin **354** was the intermediate for the reaction (Figure 101).

Sakurai et al., using malononitrile (**91**) with salicylaldehydes and acetophenones **87** in the presence of ammonium acetate, also synthesized benzopyrido [3,4-c]coumarinimides **360a**–**e** and benzopyranopyridopirimidines **359a**–**g** [186]. The 3-Amidinocoumarinimide (**A**), analogous to the abovementioned **354**, was the proposed intermediate for these reactions (Figure 102).

In 1987, O’Callaghan studied the reaction of salicylaldehydes with alkyl acetoacetate and excess of ammonia in acetic acid at room temperature, which yielded the dihydropyridine derivatives **365a**–**h** and their zwitterions **364a**–**d**. Zwitterions in solution changed slowly to the hydroxy derivatives [187]. Mild oxidation of them with 2 N HNO_3_ resulted in pyrido [3,2-*c*]coumarin derivatives **367a**–**e**. The same derivatives were obtained by the reaction of salicylaldehydes with alkyl 3-aminocrotonate (Figure 103).

Navarrete-Encina et al., performed, also, the reaction of salicylaldehydes with ethyl 3-aminocrotonate (**368a**) to get pyrido [3,4-*c*]coumarin **371a**,**b** or pyrido [5,6-*c*]coumarin derivatives **372a**–**l** [188]. Using acetic acid in ethanol they obtained dihydropyrido derivatives **370a**,**b**, which upon oxidation with CrO_3_ led to **371a**,**b**. With glacial acetic acid and heating, they synthesized the pyridocoumarins **372a**–**l**. In the proposed mechanism, a condensation of carbonyl group of salicylaldehyde (**51w**) with the 3-aminocrotonate **368a** gave the intermediate **A**, which upon cyclization led to coumarin intermediate imine **B** (Figure 104). In glacial acetic acid 1,4-addition of amine group of **368a** to the coumarin **B**, followed by intramolecular cyclization of the isomer **D** of intermediate **C** to E and elimination of ammonia afforded to the dihydropyridine moiety F. Oxidation of the latter furnished the pyridocoumarin **372a**. Upon 1,4-addition of the α-carbon of 3-aminocrotonate to **B** in acetic acid/ethanol with decreased acidity the intermediate **G** was formed. Intramolecular cyclization of its isomer **H** to tetrahydropyridine **I**, followed by elimination of ammonia from intermediate **J**, led to dihydropyridocoumarin **370a**.

In 2011, Magedov and coworkers utilized a multi-component reaction of salicylaldehydes with 3-aminopyrazol-5-ones **373a**–**c** and ethyl acetoacetate for the formation of 2,3-dihydrochromeno [4,3-*d*]pyrazolo [3,4-*b*]pyridine-1,6-diones **374a**–**o,** in order to study their antibacterial properties [189]. They synthesized, also, fused pyrido [3,4-c]coumarin derivatives **375**, **379** and the pyrimidino [3,4-c]coumarin **377** using as heterocyclic amines 3-amino-5-methyl-2-phenylpyrazole (**187a**), 6-aminouracil (**378**) and 3-amino [1,2,4]triazole (**376**), respectively (Figure 105). In the proposed mechanism, 3-acetylcoumarin (**380**) was formed in situ from salicylaldehyde and ethyl acetoacetate followed by a condensation with aminopyrazole. The compounds were tested for their antibacterial activity against gram(+)-strains. Compounds **374i** and **374o** inhibited the growth of *Staphylococcus epidermidis* with MIC = 6.3 and 25 μM, respectively. Derivative **374i** was also active against methicillin-resistant *Staphylococcus aureus*, inhibiting the growth of this pathogen with MIC = 25 μM.

Gomha and Riyadh reported analogous reactions of salicylaldehyde (**51w**) and ethyl acetoacate (**90k**) with heterocycle amines such as **381a**–**g**, **385a**–**c** and **387a**,**b** [190]. They had isolated the intermediate 3-acetylcoumarin (**380**), which reacted further with pyrimidine derivative **383** to give the fused compound **384** (Figure 106).

Palacios et al., synthesized the pyrido [3,2-*c*]coumarin **395** by oxidation of 5*H*-benzopyrano [4,3-*b*]pyridine **394** with ruthenium chloride in the presence of sodium periodate [191]. A Diels–Alder reaction of allyloxy-derivative **391** or propargyloxy-derivative **393** furnished the compound **394** (Figure 107). Derivatives **391** and **393** were obtained by a Wittig reaction of aza-ylide **389** with the derivatives of salicylaldehyde **390** and **392**, respectively.

Keskin and Balci used an oxidation with CrO_3_ for the synthesis of pyrido [3,2-*c*]coumarins **399a**–**f** from the chromenopyridines **398a**–**f** [192]. A [4 + 2] Diels–Alder cycloaddition reaction of the adduct **A** from the reaction of 2-propargyloxybenzaldehyde (**392**) with propargylamine (**397**) resulted in **398a** (Figure 108). A base-catalyzed isomerization of the terminal alkyne of the imine moiety by DBU resulted to the intermediate allene **B**, followed by a Diels–Alder reaction to the dihydropyridine intermediate **C**. 1,5-H shift of the latter afforded **398a**.

## 3. Conclusions

From the literature review, pyridocoumarins, naturally occurring or synthetic, were found to have interesting biological activities. The synthetic strategies for the synthesis of pyridocoumarins involve two main routes. The formation of the pyridine ring in one route is achieved from a coumarin derivative, such as aminocoumarins, hydroxycoumarins, or other coumarins. In the other route, the pyranone moiety is formed from an existing pyridine or piperidine or phenol derivative. [4 + 2] Cycloaddition reactions, multi-component reactions (MCR), as well as metal-catalyzed reactions are useful for the above syntheses. Name reactions, such as Skraup, Skraup–Doebner–von Miller, Povarov, Friedlander, and Krohnke, are useful for these syntheses.

Pyridocoumarins present anti-cancer, anti-HIV, antimalarial, analgesic, antidiabetic, antibacterial, antifungal, antioxidant, and anti-inflammatory activities. Especially, pyrido [3,4-*c*]coumarin **302** presented good antimalarial activity. Some pyrido [3,2-*c*]coumarins had better antitumor activity than 5-fluorouracile, while dihydrochromeno [4,3-*b*]quinoline derivatives exhibited very good antitumor activity. Pyrido [3,2-*c*]coumarin and pyrido [3,4-*c*]coumarin derivatives showed antibacterial and antifungal activities. Pyrano [2,3-*f*]quinolin-2-one **97** had excellent anti-HIV activity.

We hope that this review will benefit researchers, not only in the field of pyridocoumarin derivatives, but generally, in the area of coumarins.

## Data Availability

Not applicable.

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
