# Peer review of "An Overview on the Synthesis of Fused Pyridocoumarins with Biological Interest"

_molecules, 2022, doi:10.3390/molecules27217256_

Round 1
Reviewer 1 Report
The review by Douka and Litinas, entitled: “An Overview on the Synthesis of Fused Pyridocoumarins with Biological Interest”, is an excellent account that summarizes the chemistry and pharmacological properties of pyridocoumarins, a class of synthetic and naturally occurring organic molecules with vast and interesting biological activities, such as anti-cancer, anti-HIV, antimalarial, analgesic, antidiabetic, antibacterial, antifungal, 15 anti-inflammatory, antioxidant activities. Particularly, this long (94 pages), with 193 references, review is dedicated to the synthetic strategies for the synthesis of pyridocoumarins and presents the biological properties of those ligands. Most of the synthetic strategies involve the formation of the pyridine ring from a coumarin derivative. Next, the formation of pyranone moiety follows from an existing pyridine, piperidine or phenol intermediate. [4 + 2] Cycloaddition reactions, multicomponent reactions, as well as metal-catalyzed reactions are useful for the above syntheses. To sum up, this is a very well written review, of interest for people reading Molecules, and consequently I strongly suggest to accept it for publication as it is.
Reviewer 2 Report
This review article “An Overview on the Synthesis of Fused Pyridocoumarins with Biological Interest” focused on the synthesis of fused pyridocoumarins, as well as it's biological activities. The authors, themselves work on the synthesis of coumarin derivatives. Based on their own practices, they did wonderful literature-digging work. Numerous endeavors have been devoted to this work. More than 100 schemes were drawn and about 200 references were cited in the review, along with all the reaction conditions, reagents, and even the substrate scopes depicted in detail. It is a comprehensive and conclusive review that will be very beneficial to the researchers on the coumarins area.
There are some minor issues that should be addressed.
Keywords, line 18, “4+2” should be “[4+2]”.
Line 30, “was” should be “were”.
Page 3, scheme 2: “gl.” Should be defined. Or in this case, delete them is a good choice.
Page 8, line 162, “A few years later”; Page 10, line 182, “The same group earlier”; Page 11, line 191, “a little earlier”. Please replace them with specific years.
Scheme 7 and scheme 8, the catalyst amount should be in parenthesis, e.g. “Yb(OTf)3 (5 mol%)”.
Scheme 11, intermediate B and C, there are extra “-” on the phenyl group. Same as scheme 14.
Scheme 12, there are two intermediate B named.
Scheme 13, “700-800C” should be “70-80 oC”. There are two 66 named.
Line 242, “27” should be “32”.
Scheme 17, “PhMe” please keep the same format as “toluene”.
Line 284, “72a” should be “74a”.
Scheme 23 and line 353, “1,3-H shift” should be “1,5-H shift”.
Line 373, “100” should be “102”.
Line 383, please double check “Propargylamino-comound 117 ”. Line 384, “59c.” should be “59c”.
Scheme 26, please double check compound 112.
Line 475, “55a” should be “51a”.
Scheme 37 and scheme 39, the conditions are very confusing.
Scheme 38, there is no description for it in the context. Please double-check and have it revised.
Line 575, “120a” should be “115a”.
Line 585, the sentence needs to be polished.
Line 587, “175” should be “177”.
Line 590, please double-check 177 or 159.
Scheme 45, compound 177 should be 179.
Scheme 50, “1,3H-shift” should be “1,3-H shift”. Same as scheme 52.
Scheme 52, “)))” should be “ultrasonic” and “190n” should be “188n”.
Scheme 59, “166b” should be “181b”.
Line 794, please polish the sentence.
Scheme 68, “205” should be “218”. Please remove the lone pair on 219 and 220.
Line 932, “242” should be “244”.
Scheme 80, “167” should be “267”.
Line 1026, “151a” should be “251a”.
Scheme 83, “ultrasonic”, “(((”, and “sonication” (scheme 84) please keep them uniform.
Line 1045, line 1062, and line 1065, “151a” should be “251a”.
Line 1063, “278” should be “272b”.
Scheme 92, the structure is incorrect from compound 306. Please double-check.
Line 1194, “338” should be “339”.
Line 1249, “365” should be “364”.
Line 1269, “270a” should be “370a”.
Line 1307, “4+2” should be “[4+2]”.

Reviewer 3 Report
Dear Author and Editor
The paper is based on several routes of synthesis of some coumarin derivatives,
The paper is well written, the chemical pathways and reactions are very clear
I ask the author to review the scientific names of bacteria and fungi, as the name of the epithet must be lowercase
“Escherichia coli, Salmonella typhi, Staphylococcus Aureus and Bacillus Subtilis. Page 827
Salmonella typhimurium, Antifungal activity was tested against Aspergillus Niger, Aspergil page 828
lus Flavus, Penicillium chrysogenum, and Fusarium moniliforme page 829
BEst wishes
